

# Evolution of the scholarly mega-journal, 2006–2017

Bo-Christer Björk

Information Systems Science, Hanken School of Economics, Helsinki, Finland

## ABSTRACT

Mega-journals are a new kind of scholarly journal made possible by electronic publishing. They are open access (OA) and funded by charges, which authors pay for the publishing services. What distinguishes mega-journals from other OA journals is, in particular, a peer review focusing only on scientific trustworthiness. The journals can easily publish thousands of articles per year and there is no need to filter articles due to restricted slots in the publishing schedule. This study updates some earlier longitudinal studies of the evolution of mega-journals and their publication volumes. After very rapid growth in 2010–2013, the increase in overall article volumes has slowed down. Mega-journals are also increasingly dependent for sustained growth on Chinese authors, who now contribute 25% of all articles in such journals. There has also been an internal shift in market shares. PLOS ONE, which totally dominated mega-journal publishing in the early years, currently publishes around one-third of all articles. Scientific Reports has grown rapidly since 2014 and is now the biggest journal.

## INTRODUCTION

Electronic dissemination on the web has long been envisioned as a "game changer" for the publishing of scholarly peer reviewed journals. Open access publishing, in which readers no longer pay for access content, has become possible due to this (*Suber, 2012*). Nevertheless, leading mainstream publishers have been slow in adapting OA, simply because the subscription model is still very lucrative (*Björk, 2017a*). The big change in their business model has been from paper to electronic delivery, and the bundling of journals into huge e-licenses.

The leading publishers (commercial, society and university press) have consequently been quite cautious in starting new OA journals or converting existing journals to open access funded by author-side payments. They have instead partially opened around 10,000 journals in a hybrid form, in which authors can pay to make their articles open in otherwise closed subscription journals (*Laakso & Björk, 2016*).

Electronic web delivery has also made possible experiments with new types of peer review, for instance open peer review, in which the manuscripts have been posted on the web and readers provide reviews. One type of review which web publishing indirectly has facilitated is a review based only on scientific soundness, not on the perceived importance

Corresponding author
Bo-Christer Björk,
Bo-Christer.Bjork@hanken.fi

of the findings. The reason is that electronic-only OA journals no longer need to restrict the number of articles yearly published, but can easily scale up according to the number of submissions.

Using such review methods several publishers have started so-called mega-journals, in the wake of the phenomenal success of the pioneering PLOS ONE journal. Over the last ten years the total article output of such journals has rapidly grown, and nowadays constitutes a significant share of all output in OA journals.

A number of authors have proposed slightly varying definitions of what constitutes a mega-journal (*Norman, 2012*; *Spezi et al., 2017*). The definition of a mega-journal used in this study is the same as in *Björk (2015)*. A mega-journal has to fulfil five primary criteria:

- A big publishing volume or aiming at it
- Peer review by scientific soundness only
- Broad subject area
- Full open access
- Funded by authors paying publishing fees.

In addition, a mega-journal should fulfil several (but not necessarily all) of a number of secondary criteria. These include:

- Rapid publication
- Moderate author fee
- High prestige publisher.

For the full list see (*Björk, 2015*).

The aim of this study was to provide new updated data on the longitudinal evolution of mega-journal output, and to compare that with the article volumes of related phenomena such as articles in full OA journals and hybrid OA journals.

## EARLIER RESEARCH

There have been relatively few empirical studies of mega-journals. In addition, there have been a number of interesting newsletter and blog items, which to some extent have reported data, but which also express opinions about the phenomenon. In addition to the more scholarly studies, advocates for OA and sceptics have debated the merits and dangers of mega-journals. Titles such as: *"Open Access Megajournals—have they changed everything"* (*Binfield, 2013*), "and *"Mega-journals: the future, as stepping stone to it or a leap into the abyss?"* (*Pinfield, 2016*) describe the discourse pretty well.

Academic studies concerning mega-journals have covered a number of aspects. Topics which have been covered in the earlier literature include:

- Definition of a mega-journal, features, lists of journals (*Norman, 2012*; *Björk, 2015*; *Spezi et al., 2017*).
- Bibliometric studies of citations, etc. (*Björk & Catani, 2016*; *Wakeling et al., 2016*).
- Author surveys, factors affecting journal choice, etc. (*Solomon, 2013*).
- Case studies of individual journals (*Wilson & Humphrey, 2017*; *Wakeling et al., 2017*).

*Spezi et al. (2017)* provides an excellent review of the literature to date, and the reader is referred to that article for a more in-depth discussion.

There have been a handful of studies and blog-items that, in particular, have included data on the article volume development of mega-journals (*Binfield, 2013*; *Björk, 2015*; *Spezi et al., 2017*). The range of included journals varied somewhat but since all tend to have included the leading journals PLOS ONE and Scientific Reports, they are roughly comparable. All these show a very rapid growth period from 2010, which seems to have started levelling out in 2013–2015.

## METHODS

The basis for the list of mega-journals studied were the 14 journals which had been identified in the earlier study (*Björk, 2015*). Springer Plus was included, despite the fact that it has ceased publishing from the start of 2017. In addition, five additional journals were added. Medicine, F1000 research, and BMC research notes had been included in the previous study of *Spezi et al. (2017)*. Heliyon started publishing only recently. The Cogent series of 15 mega-journals (Cogent Engineering, Cogent Social Sciences, etc.) were also added and considered as one journal.

The publication volumes for the journals were checked 15-16.1.2018 using Scopus for all the journals included in that index, using the advanced search function which allows searching articles in a particular journal. Only articles were included and all other types of indexed items (reviews, errata, retractions) were excluded. In the case of the Cogent series of journals and the Journal of Engineering, articles were hand counted from the websites.

The share of Chinese authors in the journals was obtained using the Scopus numerical breakdown of country affiliation of the authors. Thus the count is based on where the author is working, not directly on nationality. What Scopus counts are articles with at least one author from the country in question. Since many articles have more than one author, the sum of the country affiliations will be higher than the number of articles (had only the corresponding authors been counted the sums would be equal). This is not a problem if the longitudinal changes in shares are studied or in comparisons between countries or with other disciplines. The same method has been used in the earlier study by *Wakeling et al. (2016)*.

## RESULTS

### Longitudinal development

From a longitudinal perspective the evolution of mega-journals can be split into a number of major phases. During the first phase, PLOS ONE was the one and only of its kind and grew from 138 articles in 2006 to 6,864 in 2009. When its success started to be apparent, several other established mainstream publishers launched their own mega-journals. Nine of the journals in this study were launched in either 2011 or 2012 and the period up to 2013 saw a rapid growth in combined output. From 2015 onwards the major developments have been that Scientific Reports has caught up with and surpassed PLOS ONE in article

**Table 1  Development of article volumes in mega-journals 2010–2017.**

| Journal: | Number of published research articles | | | | | | | |
|---|---|---|---|---|---|---|---|---|
| | **2010** | **2011** | **2012** | **2013** | **2014** | **2015** | **2016** | **2017** |
| **"Big Two":** | | | | | | | | |
| Scientific Reports | | 208 | 819 | 2,498 | 3,940 | 10,707 | 20,358 | 24,077 |
| PLOS ONE | 6,864 | 13,701 | 23,426 | 31,404 | 30,398 | 27,858 | 21,770 | 20,098 |
| **Converted journals:** | | | | | | | | |
| Medicine | | | (22) | (29) | 296 | 1,814 | 2,844 | 2,761 |
| **Middle tier:** | | | | | | | | |
| Springer Plus | | | 77 | 666 | 743 | 881 | 2,011 | 0 |
| IEEE Access | | | | 62 | 118 | 230 | 758 | 2,070 |
| BMJ Open | | 98 | 625 | 894 | 1,059 | 1,292 | 1,735 | 1,683 |
| Cogent Series | | | | | 110 | 516 | 1,298 | 1,432 |
| AIP Advances | | 251 | 373 | 396 | 558 | 930 | 1,240 | 1,395 |
| PeerJ | | | | 229 | 474 | 826 | 1,309 | 1,367 |
| **Smaller journals:** | | | | | | | | |
| BMC Research Notes | 343 | 544 | 673 | 532 | 958 | 870 | 526 | 739 |
| Royal Society Open Science | | | | | 50 | 246 | 414 | 648 |
| G3 | | 63 | 167 | 249 | 418 | 323 | 285 | 352 |
| F1000 Research | | | 42 | 204 | 269 | 200 | 421 | 325 |
| Sage Open | | 46 | 116 | 222 | 326 | 288 | 367 | 304 |
| Heliyon | | | | | | 29 | 156 | 249 |
| Biology Open | | | 140 | 160 | 137 | 183 | 217 | 218 |
| FEBS Open Bio | | 4 | 52 | 78 | 121 | 110 | 118 | 170 |
| Journal of Engineering | | | | 20 | 102 | 80 | 69 | 92 |
| Elementa, Science of the Antropocene | | | | 12 | 12 | 39 | 52 | 27 |
| **ALL MEGAJOURNALS** | **7,207** | **14,915** | **26,510** | **37,626** | **40,089** | **47,422** | **55,948** | **58,007** |
| Big two (2) | 6,864 | 13,909 | 24,245 | 33,902 | 34,338 | 38,565 | 42,128 | 44,175 |
| Converted (1) | | | | | 296 | 1,814 | 2,844 | 2,761 |
| Middle tier (6) | | 349 | 1,075 | 2,247 | 3,062 | 4,675 | 8,351 | 7,947 |
| Smaller journals (10) | | 685 | 1,269 | 1,472 | 2,241 | 2,308 | 2,781 | 3,158 |

numbers, and that many of the middle tier journals have consolidated their position. The overall development is shown in Table 1.

The journals can be grouped into four groups, each with its own characteristics. The first one consists of PLOS ONE and Scientific Reports, which each contribute around one third of all mega-journal articles. The second group consists of the single subscription journal Medicine, which converted to an OA mega-journal in 2014. The article volume prior to conversion in 2012–2013 is shown in parenthesis, and demonstrates a staggering hundredfold growth in just a couple of years.

The third group contains six journals with between 1,000–2,000 articles per annum. Of these, three are from highly reputable society publishers with portfolios of several journals (BMJ, AIP and IEEE). PeerJ is a start-up with no prior publisher brand name to leverage.

**Table 2  Share of authors with affiliation in China in mega-journal publication volumes.**

| Journal: | Share of authors with an affiliation in China (%) | | | | |
|---|---|---|---|---|---|
| | 2013 | 2014 | 2015 | 2016 | 2017 |
| IEEE Access | 6 | 14 | 24 | 47 | 55 |
| Medicine | 0 | 28 | 37 | 39 | 54 |
| AIP Advances | 32 | 42 | 40 | 40 | 40 |
| Scientific Reports | 29 | 39 | 39 | 37 | 31 |
| FEBS Open Bio | 6 | 6 | 19 | 26 | 30 |
| Biology Open | 3 | 1 | 6 | 15 | 20 |
| PLOS ONE | 19 | 20 | 19 | 16 | 16 |
| PeerJ | 2 | 4 | 7 | 15 | 16 |
| Royal Society Open Science | | 2 | 3 | 3 | 8 |
| G3 | 6 | 5 | 7 | 11 | 7 |
| BMJ Open | 4 | 7 | 7 | 7 | 7 |
| Heliyon | | | 14 | 9 | 5 |
| BMC Research Notes | 2 | 2 | 1 | 2 | 2 |
| F1000 Research | 2 | 2 | 2 | 2 | 2 |
| SAGE Open | 0 | 1 | 1 | 1 | 1 |
| Elementa, Science of the Antropocene | 8 | 0 | 0 | 4 | 0 |
| **In all journals** | **18** | **21** | **23** | **25** | **25** |

Springer Plus also belonged to this category before the journal stopped publishing in 2017. Cogent has preferred to split, what in this study is regarded as one mega-journal, into 15 distinct journals together covering all sciences. The journals in this middle tier contribute 14% of all articles.

The fourth and last group includes the remaining 10 journals, with predicted journal volumes of clearly less than 1,000 in 2017. Summed up they only publish 5% of all articles. In this group there is one journal, which is concentrating only the social sciences and humanities; Sage Open. Only two of the journals in the group have so far JCR impact factors.

## Rising share of Chinese authors

There has been a shift in the origin of authors who publish in mega-journals. Of particular interest is the high proportion in some of the biggest journals of authors affiliated with Chinese universities or institutes. Already *Wakeling et al. (2016)* in their bibliometric analysis noted a Chinese share of around 40% in both Scientific Reports, AIP Advances and Medicine. For this study Chinese author shares for the same journals and some additional journals were estimated from 2013 to 2017 using the Scopus index search facility. The results are shown in Table 2. As a comparison point the overall percentage of China based authors of all Scopus articles was 16% in 2013 and 20% in 2017.

The distribution over the journals is highly skewed, two journals having more than half Chinese authors, and five over 30%. Overall the share has risen but seems to have stabilised around 25%.

## DISCUSSION

The article output of mega-journals should be seen in context, for instance as part of the publications from all credible peer reviewed journals (so-called predatory OA journals excluded). A good tool for measuring this is the Scopus index which currently indexes around 20,700 mostly English language journals, including 17 of the 19 journals in this study.

Between 2010 and 2016 the overall number of articles indexed in Scopus grew by 28%, to around 2,170,000. In a separate on-going study together with Mikael Laakso, we have estimated that in 2010 the share of OA articles of all Scopus articles was 10.3% and grew to 19.4% by 2016. The by far biggest growth rates in this period were for mega-journals (0.4% to 2.6% Scopus share) and articles in hybrid OA journals (0.6% to 2.0%). The numbers for hybrid journals are from a recent separate study (*Björk, 2017b*). In 2010 almost every second OA article was still published in a journal not charging authors (81,000 vs 93,000 in APC-charging) but by 2016 charging the authors had started dominating the picture (129,000 free vs 293,000 for which APCs were paid).

This study is a straightforward empirical study using robust data available from high-quality indexing services. No sampling has been required. It is easily replicable and can also be renewed at a later stage to study subsequent developments. The results for the earlier years are well in line with the results from research reported in the "earlier research" section. The minor differences can be explained by slightly different lists of included journals and using Scopus vs counting articles from journal websites. A key challenge is obviously also in future studies to identify new Mega-journals as such are started up or converted from subscription journals.

A very challenging future research topic is what the effects of the "scientific soundness" only review criterion has on the internal citation patterns of articles in mega-journals vs. traditional journals (*Björk & Catani, 2016*; *Wakeling et al., 2016*).

## CONCLUSIONS

All in all, the developments in article numbers indicate that mega-journals have found a place in scholarly publishing. From a business perspective they complement well the journal portfolios of major commercial and society publishers, and thrive in symbiosis with more selective journals, for instance via rejected submissions being redirected to them via so-called cascading reviews (*Spezi et al., 2017*). Mega-journals will not revolutionize the industry and the way mainstream peer review works, but they cater to the needs of particular groups of authors in providing rapid publication, better predictability of getting a submission accepted and reasonable brand recognition in publication lists.

### Funding

The author received no funding for this work.

## Competing Interests

The author declares that there are no competing interests.

## Author Contributions

- Bo-Christer Björk conceived and designed the experiments, performed the experiments, analyzed the data, contributed reagents/materials/analysis tools, wrote the paper, prepared figures and/or tables, reviewed drafts of the paper.

## Data Availability

The raw data is included in the tables in the manuscript.

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
