# Peer review of "Evolution of the scholarly mega-journal, 2006–2017"

_PeerJ, doi:10.7717/peerj.4357_

## Round 0.1 · original submission · Major Revisions

As you can see, the reviewers have raised a few concerns about your methodology. In particular, data gathering and analysis have raised questions that need to be addressed.

·

Basic reporting

The paper is well written and meets all journal standards for clarity etc.
Some minor points:
Lines 224-225. “In a separate on-going study together with Mikael Laakso, we’ve estimated that there were 327,000 such articles in Scopus in 2016.” It’s not clear to me from the preceding paragraph what “such articles” refers to. The number seems too high to be mega-journal articles. Please clarify.

On line 181 the reference is a number, rather than the author name.

Lines 230-232. “Taking that as a rather conservative estimate for the current APC-funded article share, the estimate for articles for which APCs were paid is around 160,000 articles. These estimates are presented in table 4. Around 2,9 % of articles were published in mega-journals.”
It took me a couple of reads to realise that the "2.9% of articles" referred to ALL articles published, and not of articles for which an APC was paid (which is what the previous sentence refers to). I think this paragraph could be restructured to make it clearer.

I think all of the Tables would benefit from some additional thought regarding layout:

In Table 1, for completeness the SUM row should include the data from 2006-2008.

I think Table 2 would be clearer if it showed the increases as percentages rather than factors.

In Table 3, the Chinese share of all mega-journal articles (%) is shown in the far right hand columns, under the raw numbers – why not put them in the left hand columns under the other percentages?

I find the layout of Table 4 a little strange – I’m not sure why different (untitled) columns are used for the various article counts. I think this could helpfully be compacted.

Experimental design

I have two significant concerns.

The first is the decision to class RSC Advances and Palgrave Communication as mega-journals. RSC Advances has the following criteria for publication on their website (http://www.rsc.org/journals-books-databases/about-journals/rsc-advances/):
“The criteria for publication are that the work must be high quality, well conducted and advance the development of the field. Articles submitted to the journal are evaluated by international referees for the overall quality and accuracy of the science presented.”

Similarly, Palgrave Communications state the following as their aims and scope (https://www.nature.com/palcomms/about):
“The journal editors are dedicated to publishing high-quality original scholarship. Actively welcomed for submission is research on agenda-setting issues, grand societal challenges and emerging areas of thinking, irrespective of the field of study. This also includes research that reflects on, or seeks to inform, policymaking of all types.
Palgrave Communications additionally welcomes interdisciplinary research that makes an explicit and valuable contribution to the advancement of the humanities and/or social sciences.”

While I would agree that in both cases the term “quality” leaves some room for interpretation, neither journal explicitly states (as all the other journals on your mega-journal list do) that they review solely for scientific soundness. I therefore see no difference between RSC Advances and other large, broad scope, OA journals such as Nature Communications, Science Advances, or eLife.

My view is that either these two journals should be excluded from the analysis (which I think is preferable), or instead explicit discussion of their peer review processes should be included in the paper, along with clear justification for their being considered mega-journals.

My second issue is with the method used to obtain data relating to the proportion of Chinese authors.

The method states that “The share of Chinese authors in some of the journals was estimated by picking 100 sequentially ordered articles published in the summer of 2017 or 2013, and by visually identifying articles were [sic] all the leading authors were Chinese in the tables of contents.”

I think this needs clarification, as there seem to be a number of questions left unanswered. Why was summer chosen as the sampling period? Was only the nationality of the lead author considered? How was the nationality of the author determined (just based on name? Institutional affiliation?).

I also have an issue with the sample size, particularly in the case of PLOS ONE and Scientific Reports. Given the publishing volume of these journals, the sample required to estimate a proportion with a reasonable degree of confidence is significantly higher than 100.

Given this it’s unclear to me why this sampling method was favoured over the use of a bibliometric database (e.g. Scopus, which was used for data collection later in the paper, and was used by Wakeling et al. to generate the 2015 figures you include in the table). Using Scopus would have allowed you to obtain true figures rather than estimates.

Out of curiosity I ran several such searches in Scopus, and in some cases found quite different results to the ones presented in the paper.
e.g.
For 2013, Scopus shows 18.5% of PLOS ONE articles having an author affiliated with a Chinese institution (paper suggests 12%). For Scientific Reports the figure is 29% (compared to 25%). For AIP Advances Scopus shows 32% (compared to 44%).

To be clear I suspect that were the full analysis done in Scopus, the conclusions would be broadly similar. But the Scopus approach is much more technically sound.

Validity of the findings

I am confused by some of the data presented in Table 1. The outputs of RSC Advances and Medicine prior to their conversion are shown as 99 each year. I suspect these figures may be place-holders that have not been updated – RSC Advances published many thousands of articles in those years, while Medicine published fewer than 99 each year prior to 2014.

The potential link between JIF and submission rates is interesting, but as the author notes, establishing cause and effect here is difficult. Almost all mega-journals (PLOS ONE being the notable exception) show year-on-year growth between 2011 and 2016. I think therefore that if the paper wishes to provide evidence of a potential link between Impact Factor and submission rates, it needs to put post-JIF submission increases in context (i.e. identify whether post-JIF growth is at a faster rate to pre-JIF growth).

To illustrate this it is interesting to note that the data presented in Table 1 shows the one and two year increases in PLOS ONE articles following their receiving a JIF in 2010 (2.05 & 3.48) and almost identical to the increases in the two years following 2007 (2.16 & 3.50). Indeed PeerJ actually grew more slowly in the years after it got an Impact Factor (1.67 & 1.82) than it in the two years after 2013 (2.03 & 3.34). I suspect that if post-JIF increases are viewed in context it may be possible to identify more clearly those journals which were most effected by the JIF.

I was also left wondering whether the author might have drilled deeper into the geographic factors he alludes to on line 181. We know that in some countries (China being the most obvious, Spain another example) publication in journals with JIFs is formally incentivised. I wonder if analysing submission rates for these countries pre- and post-JIF might reveal something?

Given the peer review instructions I feel obliged to point out that the second half of the following sentence in the conclusion is not really supported by evidence in the paper: "its also evident from the numbers, that mega-journals have established a definite niche for themselves, based on benefits to authors in combination with the reasonable brand recognition they provide in CVs"
Since the paper does not examine why authors submit to mega-journals, it is probably necessary to make it clear that the proposed reasons are based on the literature or speculation.

Comments for the author

It is helpful to see updated mega-journal publication data, and there are some interesting findings in the paper. I do feel though that some changes to the data collection are necessary to make this paper methodologically sound.

·

Basic reporting

No comment

Experimental design

Identification of Chinese authors needs more detail. What is leading author – first author, corresponding author? And is this the only author looked at in order to identify the article as Chinese one? And what is “visually identifying”? Is that checking the relevant author’s affiliation, or whether the author name looks Chinese? – In the latter case, Chinese working in non-Chinese institutions will be lumped with authors working in China, which is quite another matter. Author behaviour will be influenced by institutional surroundings, not by nationality or ethnicity per se.
A sample of 100 seems to me, intuitively, to be a bit small for the large-volume journals – could this number be expanded to 200 for some journals and years, to make the numbers more reliable?

The 2017 numbers in Table 1 are 2 times the actual numbers end of June. Finding numbers by end of September and calculating on that basis could (should?) give more reliable numbers. If it could be without delaying publication much, I would advise this be done.

No comments on other aspects.

Validity of the findings

No comment

Comments for the author

Some comments on minor points, referring to line numbers in brackets:
[12] [which authors pay for the dissemination services] I think you should enlarge on this - as it stands, it could be read as that you are of the opinion there is nothing done in the pre-dissemination phase, that authors need pay for. Which I cannot think is your opinion, based on your previous publications. (Might be I misinterpret your use of the term “dissemination” here?)

[17] Can any comparable number for Chinese authors in general in OA or TA journals be found?

[51] A tentative percentage would make this a better read.

[65] No Björk 2016 in the literature list?

[87] [5] looks like a literature reference, but according to another standard. Should it be [2017]?

[133-134] a. for [as], substitute [a]
b. No, no different font in Table 1, but suspicious-looking rows of 99’s for both converted journals.

[158] For [of], substitute [or]

[228 ff] The percentage of OA articles published with APC payment is problematic. Crawford (who, alas, publishes himself and does not submit his works to peer scrutiny) found that for 2014 that 57 per cent of articles in OA journals were paid for through APC. (See https://scholarlykitchen.sspnet.org/2015/08/26/do-most-oa-journals-not-charge-an-apc-sort-of-it-depends/). Despite lack of peer review, Crawford’s numbers generally can be trusted. Crawford analysed the whole set of DOAJ journals, while you limit your analysis to Scopus. I take for granted that what is missing in Scopus generally is small OA journals which tend to be non-commercial. That means the APC payment rate in your data should be even higher than Crawford suggest, at least 60 per cent. This is, however, not a major point of your manuscript.

I hope you follow up later with some analysis of the background for, and possible consequences, of the large influx of Chinese authors in mega-journals, and compare this to the more standard journals. Also, why only to some of the journals?

---

## Round 0.2 · Major Revisions

I was asked to take over as Editor of your article, after the previous Editor became unavailable

Thank you for the revised manuscript and addressing the reviewers comments.

An error must have occurred during the revision, as the Table 2 in the text is described to show the contribution of Chinese authors to mega OA journals, but the table at the end of the manuscript is titled Journal volume development and presents data that are not clearly described.
Also table 1 does not clearly show what the numbers present (No. of published manuscripts?) - this should be clearly stated. Also, the summary at the end of the table is not clearly labeled that it is the summary of the results above. The subheading for a journal group should also be stated above the relevant group.

Furthermore, the methodology is not clearly explained: 1. the factor used for multiplication is not explained; 2. how the origin of articles from China was determined. The methodology for estimating the overall number of Chinese authors in Scopus is not clear (in relation to the statement in the Results section, lines 137 and 138).

Finally, without the deleted tables, the manuscript has been trimmed down so much that is offers only rudimentary information, seriously restricting its methodological soundness. For example, what happened with the citations received by these journals over time? The limitations of the study have not been sufficiently addressed in the Discussion section.

There are also some typing errors, like in line 99. Please check your manuscript carefully for language and style. Also, there are statements in the manuscript which lack adequate references, such as in the Conclusions, lines 161-162.

·

Basic reporting

Good. There are a few typos (e.g. missing space in "PLOSONE" on L97, an extra word "that share of to be" on L154 etc).

Experimental design

The changes definitely make the paper more robust. I have only a few very minor points

L92-93 - A very small issue but for clarity you might report (perhaps in the table) which journal outputs were sourced from Scopus, and which from the journal website, and explain why (I'm aware that some journals are not indexed on Scopus, but some readers may not be). It is also usual to report whether you filtered Scopus results by "Document Type" - most studies like this limit results to "Article" and sometimes "Review" documents (to exclude corrections, letters etc). I don't think it's a major problem if you didn't do this, but should be clarified either way.

On lines 95-96 you write "...multiplying by a factor taking into account both scaling up to a full year and the time lag in Scopus for indexing...". For the sake of transparency and replicability I think you should say a little more about how this factor was calculated, and report the factors used in the final calculation. I may be being a little slow, but it's not obvious to me how comparing Scopus figures and journal website figures on the same day translates to an estimate of lag - wouldn't you need to look at the date of the last Scopus indexed article to calculate this?

L98-100 - Scopus "country/territory" data are determined by the country in which the author's affiliated institution is based (rather than on the actual nationality of the author). Reviewers always ask me to make this clear in the text, so I am passing that on!

Validity of the findings

All now seems sound.

Comments for the author

I am very glad you found my comments helpful, and note that all of the issues have been addressed. I think the use of Scopus data make the results much more robust. I have selected "minor revisions" only because of the few small points that might need looking at in the method section.

·

Basic reporting

No comment

Experimental design

The points I raised have been well answered by the author.

Validity of the findings

No comments

Comments for the author

I find that the points I pointed out as problematic have been well answered by the author, and find the article well publishable.

---

## Round 0.3 · accepted · Accept

Thank you for addressing all of the comments and revising the manuscript